# Genetic Mapping of Quantitative Trait Loci for End-Use Quality and Grain Minerals in Hard Red Winter Wheat

**Shuhao Yu** [1] **, Silvano O. Assanga** [2] **, Joseph M. Awika** [2] **, Amir M. H. Ibrahim** [2] **, Jackie C. Rudd** [1] **, Qingwu Xue** [1] **, Mary J. Guttieri** [3] **, Guorong Zhang** [4] **, Jason A. Baker** [1] **, Kirk E. Jessup** [1] **and Shuyu Liu** [1,*]

1   Texas A&M AgriLife Research, 6500 Amarillo Blvd W, Amarillo, TX 79106, USA;
    shuhao.yu@ag.tamu.edu (S.Y.); jackie.rudd@ag.tamu.edu (J.C.R.); qingwu.xue@ag.tamu.edu (Q.X.);
    jbaker@ag.tamu.edu (J.A.B.); kirk.jessup@ag.tamu.edu (K.E.J.)
2   Department of Soil and Crop Science, Texas A&M University, Heep Center, 370 Olsen Blvd,
    College Station, TX 77843, USA; silvan.ocheya@bayer.com (S.O.A.); joseph.awika@ag.tamu.edu (J.M.A.);
    Amir.Ibrahim@ag.tamu.edu (A.M.H.I.)
3   USDA Agricultural Research Service, Hard Winter Wheat Genetics Research Unit, 4008 Throckmorton Hall,
    Manhattan, KS 66506, USA; mary.guttieri@usda.gov
4   Agricultural Research Center-Hays, Kansas State University, 1232 240th Ave., Hays, KS 67601, USA;
    gzhang@ksu.edu
*   Correspondence: shuyu.liu@ag.tamu.edu

**Abstract:** To meet the demands of different wheat-based food products, traits related to end-use quality become indispensable components in wheat improvement. Thus, markers associated with these traits are valuable for the timely evaluation of protein content, kernel physical characteristics, and rheological properties. Hereunder, we report the mapping results of quantitative trait loci (QTLs) linked to end-use quality traits. We used a dense genetic map with 5199 SNPs from a 90K array based on a recombinant inbred line (RIL) population derived from 'CO960293-2'/'TAM 111'. The population was evaluated for flour protein concentration, kernel characteristics, dough rheological properties, and grain mineral concentrations. An inclusive composite interval mapping model for individual and across-environment QTL analyses revealed 22 consistent QTLs identified in two or more environments. Chromosomes 1A, 1B, and 1D had clustered QTLs associated with rheological parameters. *Glu-D1* loci from CO960293-2 and either low-molecular-weight glutenin subunits or gliadin loci on 1A, 1B, and 1D influenced dough mixing properties substantially, with up to 34.2% of the total phenotypic variation explained (PVE). A total of five QTLs associated with grain Cd, Co, and Mo concentrations were identified on 3B, 5A, and 7B, explaining up to 11.6% of PVE. The results provide important genetic resources towards understanding the genetic bases of end-use quality traits. Information about the novel and consistent QTLs provided solid foundations for further characterization and marker designing to assist selections for end-use quality improvements.

**Keywords:** bread wheat; quantitative trait loci; favorable allele; end-use quality; dough rheology; genetic biofortification

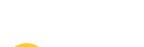



## 1. Introduction

Wheat (*Triticum aestivum* L.) is a ubiquitous cereal crop that provides caloric and nutrient requirements for humans [1]. The unique characteristics of wheat require unique processes to deliver a wide array of end-use products. In the U.S., based on market classes, different wheat has been used to produce broad-spectrum end products ranging from bread using hard wheat cultivars with high protein content and strong gluten to cakes using soft wheat cultivars with lower protein content and weaker gluten [2]. The quality of these products is contingent upon the inherent end-use quality characteristics of a given genotype. Thus, improving specific end-use quality for millers, bakers, and consumers is one of the primary components of wheat improvement.

The end-use quality in wheat can be dissected into traits such as the physical attributes of the kernels, the composition and profile of the protein fraction of the flour, the rheological properties of the dough, and the mineral-element nutritional quality, which are quantitatively inherited and influenced by environments. The kernel hardness index (HDI) is used primarily as a criterion for textural classification of grains ranging from extra soft (HDI $\leq$ 10) to extra hard (HDI > 90) based on the AACC method 55-31.01 [3]. Kernel hardness is primarily controlled by the puroindoline a (Pina) and puroindoline b (Pinb) proteins that are encoded by genes *Puroindoline a (Pina-D1)* and *Puroindoline b (Pinb-D1)*, at the hardness (*Ha*) locus on chromosome 5D [4]. The protein content is a determining factor for wheat producers to compete in grain markets and usually serves as an indicator of grain quality. Gluten proteins are the dominant grain storage proteins and constitute at least 80% of wheat flour proteins [5]. The amount and composition of three components of gluten, including high-molecular-weight glutenin subunits (HMW-GSs) encoded by *Glu-1* homoeologous genes on 1AL, 1BL, and 1DL, low-molecular-weight glutenin subunits (LMW-GSs) encoded by *Glu-3* homoeologous genes on 1AS, 1BS, and 1DS, and gliadins encoded by *Gli-1* on 1AS, 1BS, and 1DS, and *Gli-2* genes on 6A, 6B, and 6D, influence dough rheology and end-use properties [6–9]. Although concentrations are low, wheat can be an important source of trace elements such as Zn, Fe, and Se for human health [10]. In contrast, due to environmental pollution, the accumulation of heavy metals such as Cd, Pb, and As in wheat grain may cause damage to human health [11]. Efforts have been devoted to understand mineral elements' uptake and transport pathways that may relate to accumulations within the grain. A study has shown that the transportation of Zn and Cd involved the same proteins [12]. In rice and barely, the distribution of Cd involves heavy metal ATPase (HMAs) transporters [12,13]. However, the genetic mechanism which controls the mineral element concentration within grains is largely unknown.

The standard laboratory protocols for end-use quality analyses are time-consuming. For instance, a 10-g mixograph requires 8 min per sample plus an additional time for sample preparation, which depends on the operator's speed. Moreover, the amount of seed available during early generations is often insufficient for extensive end-use quality analysis. These factors are the primary reason why quality analysis is often relegated toward the advanced stages of the wheat breeding cycle, when there is a significant reduction in the number of lines to be analyzed and the amount of seed is adequate. Thus, using molecular signatures as a proxy for end-use quality can be a valuable tool for wheat improvement programs. A number of studies have been conducted which focus on identifying quantitative trait loci (QTLs) for end-use quality traits using mixographs. El-Feki et al. [14] reported QTLs for dough rheology on chromosomes 1A, 1B, 1D, 2B, 4A, 5D, 6A, 6B, 7B, and 7D. Echeverry-Solarte et al. [15] detected QTLs associated with grain protein content on chromosomes 1A, 1B, 2D, 3D, 6B, and 7B. The midline peak energy QTLs were mapped on 1B, 1D, 2D, 3D, 6B, and 7D, while midline peak time QTLs were mapped on chromosomes 1B, 1D, 2D, 3A, 5B, and 6B [15]. Dhakal et al. [2] reported that the dough rheology traits were mapped on 1A, 1B, 1D, and 7D. Little genetic information is available about wheat grain mineral element concentrations. Liu et al. [16] identified grain Zn concentration QTLs on chromosomes 1B, 2B, 3A, 3B, 3D, 4B, 5A, 6B, and 7A and QTLs associated with Fe on chromosomes 1A, 2A, 3B, 3D, 4B, 5A and 6B. Guttieri et al. [17] found the QTL associated with wheat grain Cd concentration on 5A. Understanding the genetic bases and identifying markers linked to end-use quality traits will allow breeders to better target loci within the breeding germplasm pool.

The present study used a saturated genetic map derived from a 90K Illumina iSelect array and 217 recombinant inbred lines (RILs) derived from an elite-by-elite winter wheat cross. The QTL analysis was implemented for end-use quality traits using QTL IciMapping software through the inclusive composite interval mapping function [18]. Based on this framework, the objectives of the present study were to quantify the genetic variation and map QTLs linked to kernel characteristics, dough mixing parameters, and grain mineral element concentrations in hard red winter wheat.

## 2. Materials and Methods

### 2.1. Germplasms and Field Trials

A population consisting of 217 RIL was generated by crossing between one elite line, 'CO960293-2', and a popular cultivar, 'TAM 111'. The maternal parent, CO960293-2, was developed by Colorado Agricultural Experiment Station and co-released by Colorado and Kansas Agricultural Experiment Stations [19]. The wheat streak mosaic virus resistance gene (*Wsm2*) in CO960293-2 was mapped and Kompetitive Allele Specific PCR (KASP) markers were developed [20,21]. The paternal parent, TAM 111, was developed and released by Texas A&M AgriLife Research [22]. It has excellent performance under drought stress but possesses *Glu-D1* Dx2 + Dy12, which affects bread-making. It has a glutenin to gliadin ratio of 0.79 and a high molecular to low molecular weight ratio of 0.30 [23]. The RIL plus parents were phenotyped across eight environments in a randomized block design with two replications, and samples for end-use quality analysis were drawn from the first replication of three selected environments. The selected environments were Etter, TX ($35°59'$ N, $101°59'$ W) in 2014 (ET14); Bushland, TX ($35°06'$ N, $102°27'$ W) in 2014 (BS14), and Hays, KS ($38°51'$ N, $99°20'$ W) in 2013 (HY13).

### 2.2. End-Use Quality Evaluations

Kernel characteristics, flour protein content, and dough rheological characteristics were phenotyped following the procedures described in Dhakal et al. [2]. Briefly, for each RIL, about 30-g samples from the first replication of each environment were characterized for kernel hardness index (HDI), kernel diameter (KD), and single kernel weight (SKW) using SCKS 4100 (Perten Instruments, Hagersten, Sweden). About 80 g of tempered seed samples (14% moisture content) were milled using a Brabender Quadramat Jr. Precision laboratory roller mill (Brabender Instruments, South Hackensack, NJ, USA). The characterization of flour protein content (FPC) was conducted using a real-time third generation diode array near-infrared spectroscopy (Model DA 7250, Perten Instruments, Hagersten, Sweden). Flour water absorption (WAB) and dough rheological properties were derived from a 10-g mixograph (National Manufacturing Co. Lincoln, NE, USA) based on AACC method 54–40.02 [3]. Measurements were computed at the peak, one minute before the peak (left of the peak), two minutes after the peak (right of the peak), tail (at the end of mixing), and at time_X (at the eight minutes of mixing). The variables at each level were the time, height, width, slope, and integral values. Grain samples of each genotype collected from two replications of ET14 were digested in nitric acid and hydrogen peroxide as described in Guttieri et al. [24]. Concentrations of digested samples were measured at the University of Nebraska Redox Biology Center Proteomics and Metabolomics Core Facility for arsenic (As), calcium (Ca), cadmium (Cd), cobalt (Co), copper (Cu), iron (Fe), potassium (K), lithium (Li), magnesium (Mg), manganese (Mn), molybdenum (Mo), sodium (Na), nickel (Ni), phosphorous (P), sulfur (S), selenium (Se), titanium (Ti), and zinc (Zn) elements.

### 2.3. Genotyping

The current study used a previously developed genetic map developed using RIL population and a 90K SNP array [25]. Briefly, DNA from the replicated sets of parents and RILs was extracted using the CTAB method with minor modifications [26]. The population was fingerprinted based on a hybridization-based approach of the Illumina Infinium iSelect assay (www.illumina.com, accessed on 2 January 2017). A total of 8819 polymorphic SNPs were initially used for linkage map construction using JoinMap 4.0 [27]. In JoinMap, SNPs with significant segregation distortion based on the chi-square test and those with a similarity score of 1.0 were eliminated [28]. Effectively, 5199 SNPs were used for the construction of a linkage map covering all 21 chromosomes. The physical positions of SNP were extracted according to the IWGSC RefSeq v1.0 [29] reference genome using bioinformatics tools described by Dhakal et al. [2].



*2.4. Statistical and QTL analyses*

The analysis of variance (ANOVA) for each trait across multiple environments was calculated using SAS 9.4 [30] MIXED Procedure (PROC MIXED) and the variance components of genotype (G), environment (E), and genotype-by-environment interaction (GE) were estimated using TYPE III method of moments estimation. The broad-sense heritability for end-use quality traits except mineral concentrations was calculated according to Fehr et al. [31] using the formula: $H = \sigma_G^2 / (\sigma_G^2 + \sigma_{GE}^2 / E)$, where $\sigma_G^2$ is the genotype variance, $\sigma_{GE}^2$ is the genotype-by-environment interaction variance, and E is the number of environments. The broad-sense heritability for mineral concentrations was calculated using the formula: $H = \sigma_G^2 / (\sigma_G^2 + \sigma_{GR}^2 / R)$, where $\sigma_{GR}^2$ is the genotype-by-replication interaction variance and R is the number of replications. PROC CORR in SAS was used to compute Pearson correlations for the traits across all environments.

QTL analysis was performed using QTL IciMapping software [18]. Single-trait QTL analysis was conducted in the individual environment and across environments. The genetic positions of traits were determined by the integrated composite interval mapping (ICIM) function for additive effect (ICIM-ADD) and epistasis effect (ICIM-EPI) for across-environment analyses. The threshold LOD value to declare a significant QTL for each trait was determined by 1000 permutation tests for ICIM-ADD for the individual environment and across environments. For detecting ICIM-EPI in across-environment analyses, it is too time-consuming to run permutations. Therefore, LOD = 5 was initially selected due to the computation power limit. However, the actual threshold LOD for each trait used in ICIM-EPI can be referenced from ICIM-ADD [32]. The physical position of the QTL peak was calculated based on the physical and genetic positions of the flanking markers. The designation of identified QTLs was described in Dhakal et al. [2] and Yang et al. [32].

## 3. Results

*3.1. Analysis of Variance, Heritability, and Correlations*

The means, ranges, distributions, statistical analyses, and heritability for kernel characteristics, dough rheological properties, and grain mineral element concentrations were summarized (Supplementary Table S1 and Figure S1). Generally, TAM 111 and CO960293-2 showed similar values in KD, SKW, FPC, and WAB. In contrast, CO960293-2 showed significantly higher values for HDI and mixograph traits such as MLT and MPT compared to TAM 111. The FPC ranged from 10.09 to 15.55%, with the mean at 13.00%. The mean time of dough peak formation, represented by MPT, was 4.67 min and the range of MPT was 1.50 to 8.00 min. The dough breakdown resistance indicator MPW ranged from 7.57 to 46.94%, with the mean at 24.07%. For grain mineral elements, CO960293-2 showed higher values of grain Mg, Na, P, S, K, Ca, Fe, Cu, Rb, Sr, Mo, Li, and Zn concentrations, while TAM 111 had higher values of Mn, Cd, Ni, Co, and Ti concentrations, although these values were not significantly different. Concentrations of several mineral elements were less than 1 mg kg$^{-1}$, such as Cd, Mo, As, Ni, Co, Li, and Ti (Supplemental Table S1). The average grain mineral concentrations ranged from 0.01 (As) to 5264.47 mg kg$^{-1}$ (K), and means of Cd, Fe, and Zn were 0.11, 46.07, and 44.33 mg kg$^{-1}$, respectively. ANOVA revealed significant differences ($p < 0.01$) among genotypes for all kernel characteristics and most of the mixograph traits. Only As, Cd, Co, Mo, and Ni concentrations showed significant differences ($p < 0.05$) among genotypes. Moderate to high heritability was found for most of the kernel characteristics and dough rheological traits except MTS and MTXS due to insignificant genotype effects. The highest heritability for dough rheological traits were MLT and MPT at 0.87. For grain mineral element concentrations, no heritability was found for Ca, Cu, K, Li, Mg, Mn, Na, P, S, and Rb. The heritability for Fe, Zn, Sr, Mo, As, Ni, and Ti was generally low except for Cd (0.47) and Co (0.38).

Based on overall means, high correlations were found between KD and SKW (r = 0.95), as well as WAB and FPC (r = 0.73). However, both KD and SKW were negatively and significantly correlated to FPC (r = −0.75 and −0.74) (Supplemental Table S2). The hardness index was negatively correlated with KD (r = −0.32) and SKW (r = −0.39), but positively

correlated with FPC (r = 0.20) (Supplemental Table S2). Generally, in the individual environment, KD, HDI, and SKW showed low correlations with the dough rheological traits, suggesting kernel characteristics cannot be used as predictors for the dough properties (Supplemental Tables S3–S5). Flour protein content was positively correlated with MLV, MPV, MTV, and MTXV in all three environments (r = 0.19 ~ 0.48). Most of the dough rheological traits were highly correlated, especially among the midline time, integral, width, and value traits. Additionally, high correlations were consistently found between most of the midline time and integral traits. The MLT showed a perfect correlation (r = 1.00) with MPT in all the individual environments. The MLS showed negative moderate to high correlations with MLI, MLT, MPI, and MPT but positive correlations with MTI in all three environments, suggesting that divergent selection is plausible. On the contrary, MRS showed positive correlations with MLI, MLT, MPI, and MPT because the nature of the opposite slope orientation during the dough strength build-up and breakdown stages showed consistent negative correlations between MLS and MRS in all three environments.

Substantial significant ($p < 0.05$) correlations were found among grain mineral element concentrations, except correlations between Fe with Na, P, S, Mn, Zn, Rb, and Li and correlations between Zn with K and Fe (Supplementary Table S4). Correlations among grain mineral element concentrations ranged from −0.14 (between Sr and Mn) to 0.96 (between Mg and Co). Highly significant ($p < 0.0001$) correlations were found between Cd and all other mineral elements. The correlations between grain mineral element concentrations and dough rheological traits were generally low. Grain Zn and Rb concentrations were found to be significantly correlated with MLV, MLW, MPV, MTI, and MTXI, ranging from 0.19 to 0.26. Meanwhile, significant negative correlations were found for Mg, P, and S with MRS, MTS, and MTXS, ranging from −0.14 to −0.20. Similar to dough rheological traits, correlations between grain mineral element concentrations and kernel characteristics were generally low. The correlation between FPC and As was highly significant ($p < 0.0001$). Similarly, a significant correlation ($p < 0.01$) was found between FPC and Cd at 0.21.

### 3.2. Consistent QTLs for End-Use Quality Traits

A total of 209 unique QTLs associated with 32 end-use quality traits were identified from individual or across-environment analyses (Supplemental Table S6). There were 22 QTLs associated with 12 end-use quality traits consistently detected from at least 2 individual environments (Table 1). These consistent QTLs were identified on five chromosomes, including 1A, 1B, 1D, 2B, and 3B.

**Table 1.** Consistent and pleiotropic quantitative trait loci (QTL) for kernel characteristics and dough rheological traits in 'CO960293-2'/'TAM 111' detected from at least two single environments and grain mineral element concentrations in single environments.

| QTL Name | Chr | Peak (Mbp) | Trait [a] | ENV [b] | LOD Threshold | LOD [c] | LOD (A) | LOD (AbyE) | PVE [d] | PVE (A) | PVE (AbyE) | Additive | Alleles Increase trait [e] | Pleiotropic QTL |
|---|---|---|---|---|---|---|---|---|---|---|---|---|---|---|
| *Qhdi.tamu.2B.56* | 2B | 56 | HDI | MET-ADD, BS14, ET14 | 3.2–4.6 | 9.0–23.8 | 16.6 | 7.2 | 16.7–20.3 | 12.6 | 6.2 | 1.4–2.4 | CO960293-2 | |
| *Qkd.tamu.2B.68* | 2B | 68 | KD | MET-ADD, ET14, HY13 | 3.2–8.0 | 3.7–10.0 | 4.5 | 5.5 | 6.1–8.3 | 3.7 | 4.5 | −0.01 | TAM 111 | |
| *Qfpc.tamu.3B.695* | 3B | 695 | FPC | MET-ADD, BS14, ET14 | 3.2–4.5 | 5.4–13.3 | 9.5 | 3.8 | 9.1–11.3 | 8.9 | 2.4 | −0.1–(−0.2) | TAM 111 | Y |
| *Qmpt.tamu.1A.3* | 1A | 3 | MPT | MET-ADD, BS14, ET14 | 3.2–4.6 | 8.9–12.8 | 9.2 | 3.6 | 5.2–7.1 | 3.6 | 1.7 | −0.2–(−0.4) | TAM 111 | Y |
| *Qmpt.tamu.1B.5* | 1B | 5 | MPT | MET-ADD, BS14, ET14, HY13 | 3.2–8.0 | 14.4–59.8 | 57.2 | 2.5 | 17.3–28.8 | 27.7 | 1.1 | 0.5–0.8 | CO960293-2 | Y |
| *Qmpt.tamu.1D.417* | 1D | 417 | MPT | MET-ADD, BS14, HY13 | 3.2–8.0 | 28.8–50.0 | 40.7 | 9.3 | 27.6–34.2 | 17.8 | 9.8 | 0.5–1.0 | CO960293-2 | Y |
| *Qmlt.tamu.1B.5* | 1B | 5 | MLT | MET-ADD, BS14, ET14, HY13 | 3.2–8.0 | 20.6–47.4 | 46.8 | 0.6 | 20.1–27.2 | 25.5 | 1.7 | 0.6–0.8 | CO960293-2 | Y |
| *Qmlt.tamu.1D.1* | 1D | 1 | MLT | MET-ADD, BS14, ET14 | 3.2–4.6 | 4.8–7.9 | 7.6 | 0.4 | 3.6–4.3 | 3.4 | 0.9 | −0.2–(−0.3) | TAM 111 | Y |
| *Qmli.tamu.1B.5* | 1B | 5 | MLI | MET-ADD, BS14, ET14, HY13 | 3.2–8.0 | 13.7–47.0 | 45.6 | 1.4 | 13.4–27.1 | 26.3 | 0.8 | 19.6–24.2 | CO960293-2 | Y |
| *Qmls.tamu.1B.5* | 1B | 5 | MLS | MET-ADD, BS14, ET14, HY13 | 3.2–8.0 | 10.2–15.6 | 11.2 | 4.4 | 14.2–21.9 | 14.0 | 7.9 | −1.1–(−3.2) | TAM 111 | Y |
| *Qmpi.tamu.1B.5* | 1B | 5 | MPI | MET-ADD, BS14, ET14, HY13 | 3.2–8.0 | 9.0–44.0 | 40.3 | 3.7 | 10.3–22.1 | 21.7 | 0.4 | 18.4–23.5 | CO960293-2 | Y |
| *Qmrt.tamu.1B.5* | 1B | 5 | MRT | MET-ADD, BS14, ET14 | 3.2–4.5 | 6.3–19.3 | 17.3 | 2.0 | 9.3–18.8 | 16.1 | 2.7 | 0.3–0.4 | CO960293-2 | Y |
| *Qmli.tamu.1D.413* | 1D | 413 | MLI | MET-ADD, BS14, ET14 | 3.2–4.8 | 29.2–36.2 | 12.5 | 16.7 | 17.6–31.3 | 6.3 | 11.3 | 9.6–29.4 | CO960293-2 | Y |
| *Qmtw.tamu.1D.413* | 1D | 413 | MTW | MET-ADD, BS14, ET14 | 3.2–4.5 | 9.1–10.4 | 3.3 | 5.7 | 9.7–13.6 | 3.1 | 6.6 | 0.5–2.0 | CO960293-2 | Y |
| *Qmtxw.tamu.1D.413* | 1D | 413 | MTXW | MET-ADD, BS14, ET14 | 3.2–4.6 | 7.5–10.4 | 2.8 | 4.7 | 7.4–13.6 | 2.4 | 5.1 | 0.5–2.0 | CO960293-2 | Y |
| *Qmli.tamu.1A.3* | 1A | 3 | MLI | ET14, BS14 | 3.2 | 10.7–15.8 | | | 8.4~11.1 | | | −5.4–(−17.4) | TAM 111 | Y |
| *Qmpi.tamu.1A.5* | 1A | 5 | MPI | MET-ADD, BS14, ET14 | 3.2–4.5 | 11.5–14.0 | 10.7 | 3.3 | 6.7–10.6 | 5.1 | 1.6 | −9.3–(−17.3) | TAM 111 | Y |
| *Qmrs.tamu.1B.8* | 1B | 8 | MRS | MET-ADD, BS14, ET14 | 3.2–4.5 | 5.6–12.6 | 12.1 | 0.5 | 9.5–14.7 | 11.0 | 3.7 | 0.3–0.5 | CO960293-2 | Y |
| *Qmpi.tamu.1D.66* | 1D | 66 | MPI | MET-ADD, BS14, HY13 | 3.2–8.0 | 14.6–40.7 | 29.7 | 11.0 | 22.9–30.9 | 15.4 | 8.0 | 16.2–29.7 | CO960293-2 | Y |
| *Qmpi.tamu.1D.1* | 1D | 1 | MPI | MET-ADD, BS14, ET14 | 3.2–4.5 | 5.7–6.3 | 4.4 | 1.9 | 2.9–3.9 | 2.0 | 0.8 | −5.83–(−10.5) | TAM 111 | Y |
| *Qmtw.tamu.1D.1* | 1D | 1 | MTW | MET-ADD, BS14, ET14 | 3.2–4.5 | 6.5–9.2 | 7.0 | 2.2 | 6.9–9.8 | 6.5 | 3.3 | −0.8–(−1.6) | TAM 111 | Y |
| *Qmtxw.tamu.1D.1* | 1D | 1 | MTXW | MET-ADD, BS14, ET14 | 3.2–4.6 | 6.5–7.9 | 6.2 | 1.7 | 6.9–8.9 | 5.4 | 2.5 | −0.7–(−1.6) | TAM 111 | Y |
| *Qgco.tamu.3B.32* | 3B | 32 | GCO | ET14 | 3.4 | 5.6 | | | 11.6 | | | −0.003 | TAM 111 | |
| *Qgcd.tamu.3B.46* | 3B | 46 | GCD | ET14 | 3.4 | 5.8 | | | 8.0 | | | −0.006 | TAM 111 | |
| *Qgmo.tamu.3B.540* | 3B | 540 | GMO | ET14 | 3.4 | 4.8 | | | 8.6 | | | 0.036 | CO960293-2 | |
| *Qgcd.tamu.5A.577* | 5A | 577 | GCD | ET14 | 3.4 | 5.5 | | | 8.0 | | | −0.006 | TAM 111 | |
| *Qgcd.tamu.7B.552* | 7B | 552 | GCD | ET14 | 3.4 | 5.4 | | | 7.6 | | | −0.006 | TAM 111 | |

[a] FPC, flour protein content; GCD, grain Cd concentration; GCO, grain Co concentration; GMO, grain Mo concentration; HDI, hardness index; KD, kernel diameter; MLI, midline left integral; MLS, midline left slope; MLT, midline left time; MPI, midline peak integral; MPT, midline peak time; MRS, midline right slope; MRT, midline right time; MTW, midline tail width; MTXW, midline time_X width. [b] Environments: HY13, Hays, KS, 2013; BS14, Bushland, TX, 2014; ET14, Etter, TX, 2014; MET-ADD, multi-environment trial QTL analyses for additive effects. [c] LOD, logarithm of odds; LOD(A), LOD due to additive effect; LOD(AbyE), LOD due to additive-by-environment interaction effects. [d] PVE, phenotypic variance explained; PVE (A), PVE explained by additive effect; PVP(AbyE), PVE explained by additive-by-environment interaction effect. [e] For most traits in this manuscript, except for heavy metal element concentrations, a higher value indicates better performance. The origins of favorable alleles were determined by the value of additive effects. If the additive value was positive, it suggested an origination from the maternal parent, CO960293-2, and vice versa.

### 3.2.1. QTL for Kernel Characteristics and Flour Protein Content

Eight HDI QTLs were detected on chromosomes 1A, 1B, 2B, 2D, and 5B from individual and across-environment analyses (Supplemental Table S6, Supplemental Figures S2 and S3). These QTLs explained phenotypic variations ranging from 3.4 to 20.3%. The only consistent QTL, *Qhdi.tamu.2B.56*, was identified from ET14, BS14, and across-environment analysis, explaining up to 20.3% of the phenotypical variation. The additive effect accounted for 67% of total phenotypic variations explained by *Qhdi.tamu.2B.56*. Its favorable allele from CO960293-2 increased the HDI up to 2.4%. The additive-by-environment interaction increased the HDI by 0.5% in BS14 and 0.9% in ET14 (Supplemental Table S6). Two QTLs, *Qhdi.tamu.1B.303*, which was identified from BS14 and the across-environment analyses, and *Qhdi.tamu.5B.585*, which was detected from ET14 and the across-environment analyses, increased the HDI by 1.3 and 1.2%, respectively (Supplemental Table S6).

There were nine QTLs associated with kernel diameter (KD) identified from both individual and across-environment analyses. Similar to HDI, the only consistent QTL, *Qkd.tamu.2B.68* explained 6.1 to 8.3% of phenotypical variations. Its favorable allele from TAM 111 increased KD by 0.01 mm (Table 1). Another 4 QTLs were detected on chromosomes 1D (190 Mbp), 4D (390 Mbp), 5A (610 Mbp), and 7A (581 Mbp) from the across-environment and one individual environment QTL analyses, accounting for up to 7.3% of PVE. Favorable alleles of QTLs on 1D and 5A from TAM 111 increased the KD up to 0.01 mm, while favorable alleles of QTLs on 4D and 7A from CO960293-2 also increased the KD up to 0.01 mm (Supplemental Table S6).

A total of seven QTLs associated with FPC were mapped on chromosomes 3B and 5B, explaining 3.7 to 11.3% of phenotypical variations (Supplemental Table S6, Supplemental Figures S2 and S3). One QTL, *Qfpc.tamu.3B.695*, was consistently detected, accounting for 9.1 to 11.3% of PVE. Its favorable allele from TAM 111 increased the FPC up to 0.2%, and the additive-by-environment interaction increased FPC by 0.08% in HY13 (Supplemental Table S6).

Although no consistent QTL was identified for SKW, there were 11 QTLs identified that were associated with SKW on chromosomes 2A, 2B, 3A, 5A, 6A, and 7A, explaining up to 9.1% of phenotypic variations (Supplemental Table S6). One SKW QTL detected from ET14 and the across-environment analyses, *Qskw.tamu.5A.644*, increased the SKW by 0.6 mg.

### 3.2.2. QTLs Linked to Mixograph Parameters

There were 150 QTLs identified which were associated with 23 mixograph mixing property traits on chromosomes 1A, 1B, 1D, 2B, 3B, 5B, 6D, 7B, and 7D from individual and across-environment analyses (Supplemental Table S6). Most QTLs (95.0%) for mixograph mixing properties were mapped on chromosomes 1A, 1B, 1D, and 5B. A total of 3 consistent QTLs associated with midline peak time (MPT) were identified on 1A (3 Mbp), 1B (5 Mbp), and 1D (417 Mbp) (Table 1). *Qmpt.tamu.1A.3* explained 5.2 to 7.1% of PEV. Its favorable allele from TAM 111 increased the MPT up to 0.4 min, and the additive-by-environment interaction increased MPT by 0.22 min in BS14 (Supplemental Table S6). The second consistent QTL identified from all analyses, *Qmpt.tamu.1B.5*, explained 17.3 to 28.8% of PVE with the favorable allele from CO960293-2 that increased the MPT up to 0.8 min. The additive-by-environment interaction of *Qmpt.tamu.1B.5* increased MPT by 0.14 min in HY13 (Supplemental Table S6). The third consistent QTL, *Qmpt.tamu.1D.417*, was likely *Glu-D1*, explaining 27.6 to 34.2% of PVE. Its favorable allele was from CO960293-2, which increased the MPT up to 1.0 min. Consistent with moderate to high correlations among dough rheological traits, QTLs associated with several midline time and integral traits were identified at the same genomic regions as MPT. For instance, 2 consistent QTLs associated with MLT were identified on 1B (5 Mbp) and 1D (1 Mbp). *Qmlt.tamu.1B.5* was detected in all analyses, accounting for 20.1 to 27.2% of PVE and increasing MLT by 0.8 min with the favorable allele from CO960293-2, while *Qmlt.tamu.1D.1* only increased MLT by 0.3 min with the allele from TAM 111. The additive-by-environment interaction increased the MLT

by 0.21 min in HY13. Besides MPT and MLT, QTLs associated with MLI, MLS, MPI, and MRT were also identified on 1B at 5 Mbp, explaining up to 27.1% of PVE. Its favorable allele from TAM 111 increased MLS by 3.2% $min^{-1}$. On the other hand, the favorable alleles from CO960293-2 increased MLI by 24.2% torque x min and MRT by 0.4 min. Another consistent QTL, *Qmlt.tamu.1D.1*, explained 3.6 to 4.3% of PVE. The favorable allele of *Qmlt.tamu.1D.1* was from TAM 111, while the additive effect increased the MLT by 0.3 min. Another MLT QTL on 1D near 417 Mbp was identified from one individual environment and/or the across-environment analyses, explaining 10.2 to 30.8% of total PVE with favorable alleles from CO960293-2 (Supplemental Table S6). Its additive effect explained 19.7 of the total 30.8% PVE and increased MLT by 0.5 min.

There were 3 QTLs identified on 1D at 413 Mbp associated with MLI, MTW, and MTXW, showing pleiotropic effects for highly correlated traits (r = 0.78, *p* < 0.0001). It was likely this genomic region associated with the *Glu-D1* gene explained MLI, MTW, and MTXW up to 31.3, 13.6, and 13.6% of PVE, respectively. Their favorable alleles from CO960293-2 increased the MLI, MTW, and MTXW up to 29.4, 2.0, and 2.0 percent points, respectively. Besides MPT, one consistent MLI QTL showed an association on 1A at 3 Mbp, and one consistent MPI QTL was identified on 1A at 5 Mbp. *Qmli.tamu.1A.3* was identified from BS14 and ET14, explaining 8.4 to 11.1% of PVE. The favorable allele from TAM 111 increased MLI up to 17.4%. *Qmpi.tamu.1A.5* explained up to 10.6% of PVE. Consistent with the other QTL identified in this genomic region, the favorable allele of *Qmpi.tamu.1A.5* from TAM 111 increased MPI up to 17.3% torque × min.

There were two genomic regions on 1B and 1D only associated with one consistent QTL, respectively. *Qmrs.tamu.1B.8* detected in ET14, BS14, and across-environment analyses accounted for 9.5 to 14.7% of PVE. Its favorable allele from CO960293-2 and the additive effect increased MRS by 0.5% $min^{-1}$. *Qmpi.tamu.1D.66* explained 22.9 to 30.9% of PVE in HY13, BS14, and across-environment analyses. The favorable allele of *Qmpi.tamu.1D.66* from CO960293-2 increased the MPI by up to 29.7% torque × min.

### 3.2.3. QTLs for Grain Mineral Element Concentrations

Due to the heritability of Li, Rb and Sr were 0 (Supplementary Table S1). Identified QTL associated with Li, Rb, and Sr were unreliable. Therefore, five QTLs were identified for grain concentrations of Co, Cd, and Mo on chromosomes 3B, 5A, and 7B, accounting for 7.6 to 11.6% of PVE (Table 1). There were 3 QTLs associated with Cd concentration, *Qgcd.tamu.3B.46*, *Qgcd.tamu.5B.540*, and *Qgcd.tamu.7B.552*, which accounted for 8.0, 8.0, and 7.6% of PVE, respectively. All the GCD QTLs showed negative additive effects, suggesting that alleles increased grain Cd concentration were from TAM 111. One QTL associated with Co concentration was on chromosome 3B at 32 Mbp, accounting for 11.6% of total PVE. The allele from TAM 111 increased grain Co concentration by 0.003 mg $kg^{-1}$. One QTL was identified on chromosome 3B at 540 Mbp for grain Mo concentration, accounting for 8.6% of total PVE. The favorable allele from CO960293-2 increased Mo concentration by 0.036 mg $kg^{-1}$.

### 3.3. Pleiotropic QTL

Based on QTLs identified at least twice from single and across environment analyses, nine genomic regions were found to be associated with more than one trait and thus considered to have pleiotropic effects (Figure 1). On chromosome 1A, the genomic region between 3 and 6 Mbp was clustered with QTL for MLI, MLT, MLV, MPT, MPI, MTV, MTW, MTXV, and MTXW with TAM 111 favorable alleles that explained up to 15.1% of total PVE (Supplementary Table S6). The genomic region on 1B between 3 and 8 Mbp was co-located with MLT, MLS, MLI, MPT, MPV, MPI, MRT, MRS, MRW, MRI, MTV, MTW, MTI, MTXV, MTXW, and MMST. The alleles from this genomic region that increased the phenotypic values mainly were from CO960293-2, except for MLS, MPV, MTI, MTXI, and MMST. There were three genomic regions co-located with multiple dough rheological traits on chromosome 1D. They were at 1 Mbp, between 64 and 76 Mbp, and *Glu-D1*. The favorable

alleles of the 1 Mbp region from TAM 111 explained up to 10.6% of total PVE. Favorable alleles of the other two pleiotropic regions on 1D from either TAM 111 or CO960293-2 explained up to 30.9 and 34.2% of total PVE. Two genomic regions on chromosome 5B were co-located with multiple traits. QTL for three dough rheological traits, including MLI, MPT, and MRT, were clustered at 424 Mb on 5B with CO960293-2 favorable alleles, explaining up to 5.4% of total PVE.

### 3.4. Interactions of Epistasis and Epistasis-by-Environment

There were 254 interactions of epistasis and epistasis-by-environment with a combined LOD $\geq$ 5 for all the end-use quality traits that were collected from multiple environments, except for HDI, KD, and MTXS (Supplemental Table S7; Supplemental Figure S4). However, none of these epistasis and epistasis-by-environment interactions had a combined LOD > 10. Among all the consistent QTLs, none were involved with interactions. However, there were fours QTL, *Qmlv.tamu.1B.2*, *Qmti.tamu.1B.2*, *Qmtxi.tamu.1B.2*, and *Qmri.tamu.1A.11*, that were at least detected from one individual and MET analyses and that were involved in epistasis and epistasis-by-environment interactions. The epistasis of *Qmlv.tamu.1B.2* explained 1.6% of PVE, while the epistasis-by-environment interaction explained 1.0% of total PVE. The additive-by-environment interactions of the TAM 111 allele increased the MLV by 0.6% in ET14, and epitasis-by-environment interactions between *Qmlv.tamu.1B.2* and *Qmlv.tamu.5B.581* increased MLV by 0.3 and 0.4% in HY13 and BS14, respectively. Similar results were found between the interactions involved with *Qmti.tamu.1B.2* and *Qmtxi.tamu.1B.2*. The epistasis of these two QTLs both explained 1.9% of total PVE, while the epistasis-by-environment interaction both explained 1.2% of total PVE. The epistasis between *Qmri.tamu.1A.11* and *Qmri.tamu.4A.608* explained 1.4% of total PVE, while the epistasis-by-environment interaction accounted for 0.7% of total PVE. The epistasis-by-environment interactions of the favorable allele from TAM 111 increased the MRI by 6.1% torque × min in HY13.

For MPT, there were 12 QTLs that showed epistasis interactions, with their LOD ranging from 2.8 to 6.0, while the epistasis-by-environment interactions' LOD ranged from 0.4 to 3.3 (Supplemental Table S7). The phenotypic variations explained by epistasis interactions ranged from 0.6 to 1.7%, while the phenotypic variations explained by epistasis-by-environment interactions ranged from 0.04 to 0.8%. Interactions of 2 TAM 111 alleles at *Qmpt.tamu.1D.479* and *Qmpt.tamu.2D.27* decreased the MPT by 0.19 min. However, the interaction of the TAM 111 allele of *Qmpt.tamu.1A.41* and the CO960293-2 allele of *Qmpt.tamu.6D.7* had the highest epistasis LOD of 6.0. Additionally, this interaction increased the MPT by 0.18 min, and their interaction with HY13 increased the MPT by 0.1 min.

For FPC, there were 17 epistasis interactions identified on chromosomes 1D, 3A, 4B, 5B, 6A, 6B, 6D, 7A, and 7D (Supplemental Table S7). The total LOD ranged from 5.0 to 7.6, and the PVE explained by epistasis and epistasis-by-environment interactions ranged from 2.7 to 4.4% and 1.0 to 3.2%, respectively. Three epistasis interactions, *Qfpc.tamu.2D.27* by *Qfpc.tamu.4B.597*, *Qfpc.tamu.3B.166* by *Qfpc.tamu.5B.624*, and *Qfpc.tamu.2A.770* by *Qfpc.tamu.7A.660*, had epistasis and epistasis-by-environment interactions with a combined LOD value greater than seven. The TAM 111 allele of *Qfpc.tamu.3B.166* and the CO960293-2 allele of *Qfpc.tamu.5B.624* increased the FPC by 0.1%. The epistasis-by-environment interactions increased the FPC by 0.06 and 0.04% in HY13 and BS14, respectively. The interaction between *Qfpc.tamu.2A.770* and *Qfpc.tamu.7A.660* explained the largest PVE among all the interactions and increased the FPC by 0.1% in ET14 through epistasis-by-environment interactions.

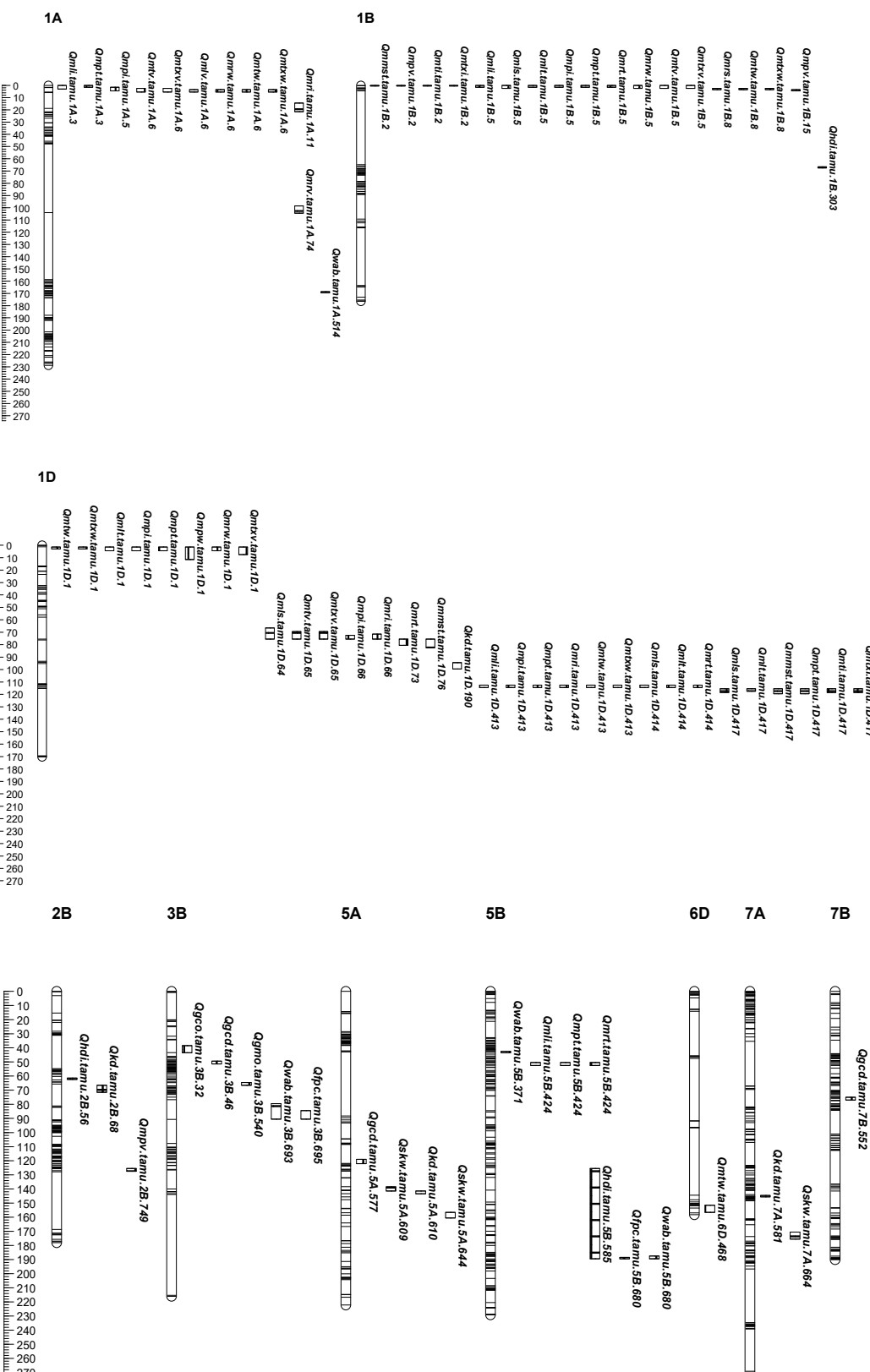

**Figure 1.** The locations of QTL peaks for end-use quality identified from individual and across-environment QTL analyses and grain mineral traits from individual environment analysis in the 'CO960293-2'/'TAM 111' RIL population. In each sub-figure, markers are represented by horizontal stripes inside a linkage group and the corresponding wheat chromosome is listed above each linkage group. Genetic distances (centiMorgans, cM) were listed on the left ruler of each sub-figure. QTLs were designated in the format as *Qtrait.tamu.chrom.Mb*.

## 4. Discussion

End-use quality analysis is an indispensable component in wheat breeding programs. However, testing and evaluating end-use quality traits are often challenged by the high cost and a large amount of grain required at early generation stages and often confounded by the environment and genetic-by-environment interactions [33,34]. Understanding the complex genetic bases of grain end-use quality traits and developing trait-associated molecular markers for marker-assisted selections can help breeders develop wheat cultivars with superior end-use quality more efficiently. Substantial efforts have been devoted to discover QTLs associated with end-use quality traits. Unfortunately, the knowledge of genetic and genomic bases of end-use quality in wheat is limited.

In the present study, we observed substantial phenotype variations among most of the end-use quality traits in this CO960293-2/TAM 111 RIL population. CO960293-2 had higher values than TAM 111 in most dough rheological traits, consistent with CO960293-2 having better end-use quality. CO960293-2 had higher grain mineral concentration values for most of the testing elements, while TAM 111 had a higher single kernel weight. This was consistent with previous findings that the mineral concentration tends to decrease when yield increases due to dilution effects [24,35]. However, the grain Cd concentration of TAM 111 was higher than CO960293-2, suggesting TAM 111 may have unique Cd uptake and distribution mechanisms.

The correlations among most of the mixograph parameters were high, especially among midline value, slope, width, integral, and time parameters, suggesting one set of mixograph parameters is enough to evaluate the dough rheology characteristics. Consistent with Tsilo et al. [34] and Dhakal et al. [2], significant correlations were observed between mixograph traits and kernel traits in individual and across-environment analyses. The results of the correlation in the present study suggest that FPC and HDI might not be a good predictor of the mixing properties of the dough. Most of the grain mineral concentrations were extensively correlated in this study. Guttieri et al. [24] reported that correlations of P with Fe and Zn were > 0.5. However, in this study, the correlations of P with Fe and Zn were low, which were similar to findings reported by Morgounov et al. [36]. Single kernel weight is a component of grain yield and is usually negatively correlated with grain mineral concentrations. Consistent with Guttieri et al. [24], Zn and S were found to be significantly negatively correlated with SKW, while Cd concentration was positively correlated with SKW.

Moderate to high heritability for all kernel characteristics and most dough rheological traits except for MTS and MTXS (Supplementary Table S1) indicated that genetic factors controlled a large part of phenotypical expressions. These results were consistent with Dhakal et al. [2] and Tsilo et al. [34], who reported a wide heritability range from 0.41 to 0.98 for mixograph parameters and kernel characteristics. In terms of grain mineral element concentrations, zero to low heritability was found for all the mineral element concentrations, except for Cd and Co, where moderate heritability was found. Guttieri et al. [24] found that the heritability of mineral elements varied by yield, ranging from low to high. The mineral elements that consistently showed moderate to high heritability were Cd and Li, whereas the heritability of Co was consistently low for trials both in higher-yielding Nebraska and and lower-yielding Oklahoma. Besides Cd, selection for improved mineral nutrients is likely affected by environmental conditions, especially under adverse conditions. [24]

In the QTL analysis, the consistent HDI QTL found on 2B near 56 Mbp were close to the QTL associated with soft durum wheat (*Triticum turgidum* subsp. *durum*) grain hardness [37]. Smith et al. [38] also found HDI QTL on 2B were likely associated with pentosans and polar lipids in the endosperm, demonstrating the contribution of grain hardness from durum wheat. Dhakal et al. [2] reported a consistent QTL for HDI on chromosome 1A near 475 Mbp, which is not far from *Qhdi.tamu.1A.520* identified in the current study that may associate with *Glu-A1*. A previously reported HDI QTL on 1D was associated with *Glu-D1*, suggesting that the gluten strength can influence grain hardness [39]. Although no previous studies showed *Glu-A1* influences grain hardness,

it is possible that the homologous locus can also influence grain hardness. It is known that the *Ha* locus and *Ha*-linked puroindoline a (Pina) and puroindoline b (Pinb) located on chromosome 5DS modulate grain hardness in wheat [40]. However, no HDI QTL was detected on 5D in this study, suggesting the diverse genetic background of bread wheat. *Qhdi.tamu.5B.585* was found on 5B near 585 Mbp, co-locating with flour protein content in a RIL population crossed between TAM 112 and TAM 111 [2]. However, no co-location of HDI and FPC QTL was observed in this study. The QTLs for SKW and KD were identified on several chromosomes, consistent with the finding that several QTLs associated with KD and SKW were identified on 5A, 6A, and 7A [41,42]. The only consistent QTL for KD, *Qkd.tamu.2B.68*, was physically close to the photoperiod response locus gene on 2B (*Ppd-B1*). Germplasms that carry photoperiod-insensitive alleles tend to flower early, enabling them to escape the late-season abiotic stresses. These also have longer times in the grain-filling stage, enabling them to reach their grain dimension potential [43]. An SKW QTL only identified in ET14, *Qskw.tamu.2B.65*, was located at the genomic region near *Ppd-B1*, conferring the high correlation between KD and SKW in this study. Consistent with a previous report, the *Ppd-B1* region was found to be associated with thousand-grain weight [44]. *Qskw.tamu.6A.608* was 4.8 Mbp away from the kernel weight QTL *QTkw.dms-6A.3* reported by Semagn et al. [41]. The genomic region on 5A near 644 Mbp showed an association with KD and SKW and was, at the same position, compared to the test weight QTL *QTwt.dms-5A.1* reported by Semagn et al. [41]. Dhakal et al. [2] reported QTLs for KD and SKW on the long arm of chromosomes 2D and 7D in TAM 111. It was unexpected that no QTL was identified on 2D and 7D in this study, since TAM 111 served as a parent for both studies.

The FPC is a quantitative trait controlled by several genes and QTLs distributed throughout the wheat genome [2,15,41,45–47]. The only consistent FPC QTL identified on chromosome 3B, *Qfpc.tamu.3B.695*, explained noticeable phenotypic variations. The protein content QTL reported by Alemu et al. [45] and Semagn et al. [41] were located at the same genomic regions but were about 100 Mbp away from *Qfpc.tamu.3B.695*, suggesting the novelty of this QTL. *Qfpc.tamu.3B.695* was very close to Endo-1,3(4)-beta-glucanase 1 at 689 Mb and was involved with seed germination and associated with grain hardness affected by protein and carbohydrate interactions [48,49]. However, no association between Endo-1,3(4)-beta-glucanase 1 and FPC was reported. Several FPC QTLs were identified on chromosome 5B, consistent with previous findings [15,41]. Although not correlated with SKW, which is considered a yield component, the FPC QTL on 5B near 680 Mbp overlapped with a yield QTL reported by Semagn et al. [41].

The dough rheological properties were often used as the direct factors to determine the wheat gluten strength that affected the end-use quality. The QTLs for dough rheological property traits were identified to be clustered on 1A, 1B, 1D, and 5B in this study, consistent with the previous findings that these traits were controlled by multiple genes or QTLs [2,34,46,50,51]. Numbers of studies have confirmed that HMW-GSs encoded by *Glu-D1*, *Glu-B1*, and *Glu-A1* influenced dough rheological properties [52–54]. In the current study, the genomic region on 1D near 413 Mbp was associated with many mixograph traits linked to the *Glu-D1* loci. Dhakal et al. [2] reported the segregation of *Glu-D1a* alleles (Dx2 + Dy12 subunit), which was inferior to *Glu-D1d* alleles (Dx5 + Dy10 subunit) in dough mixing strength [6,55]. The favorable alleles of all the QTLs of major effects associated with *Glu-D1* were from CO960293-2, consistent with the fact that CO960293-2 (*Glu-D1d*) had superior end-use quality compared to TAM 111 (*Glu-D1a*). A dough rheological trait QTL, *Qmpv.tamu.1B.558*, which was flanked by markers Bx7$^{OE}$ and IWB8798, was only detected once in the study. Bx7$^{OE}$ was linked to the 1Bx7 subunit encoded by *Glu-B1*, which has important contributions to gluten functionality [52]. No dough rheological trait QTL was found to be associated with *Glu-A1*. However, a WAB QTL, *Qwab.tamu.1A.514*, was physically close to *Glu-A1*, indicating that HMW-GSs can influence the protein concentration, which was consistent with the fact that HMW-GSs account for ~10% of total protein [56]. Three genomic regions clustered with multiple dough rheological trait QTLs were found on

the short arms of 1A, 1B, and 1D, physically close to the *Gli-A1*, *Gli-B1*, and *Gli-D1* loci that associated with γ-, δ-, and ω-gliadins or, immediately downstream LMW-GSs encoded by genes *Glu-A3*, *Glu-B3*, and *Glu-D3* [57,58]. Unsurprisingly, Dhakal et al. [2] also observed dough rheological trait QTLs clustered on the short arms of 1A, 1B, and 1D, since TAM 111 served as a parent in both studies. A genomic region on 1D near 66 Mbp was identified for a few dough rheological trait QTLs. It was likely that this region may be physically close to *Gli-D3* and had not been reported to be associated with dough rheological traits before [59].

It is estimated that more than 33% of the children and women in developing countries do not receive the Zn element [60]. Insufficient Zn intake has been correlated with the prevalence of stunting in children under age five [61]. Improving beneficial metal element accumulation in wheat is critical for maintaining good human health and has become a priority of many breeding programs. In this study, the accumulation of Zn was not consistent and no QTL was identified in this CO960293-2/TAM 111 population. Similarly, Guttieri et al. [17] could not identify QTLs associated with Zn concentration in a GWAS study, suggesting selection for grain Zn concentration likely will be ineffective and that genetic improvement for Zn is difficult to achieve within the Great Plains hard winter wheat germplasm. On the contrary, numbers of QTL associated with Zn have been identified throughout the wheat genome [10,62,63]. Molybdenum is a trace element that is essential for animals and plants. Molybdenum has been utilized explicitly in enzymes and two of them, aldehyde oxidase and sulfite oxidase, exist both in humans and plants [64,65]. Gwen et al. [66] and Wang et al. [67] reported genetic bases of Mo accumulation in rice. However, little information is available about either physiological or genetic bases of Mo accumulation in wheat. This study identified a QTL of grain Mo concentration on 3B, shedding light on molecular investigations of Mo concentration in wheat. Bhatta et al. [68] identified three marker-trait associations on wheat chromosomes 3A, 6D, and 7D for grain Co concentration based on a synthetic hexaploid wheat association panel. The QTL identified on 3B in this study is the potential novel QTL for grain Co concentration. Besides beneficial mineral elements, wheat also contains some heavy metals such as Cd, Pb, and As, which will cause prostate, lungs, and testes cancers as well as organs damages if excessively consumed [11]. Guttieri et al. [17] reported a Cd QTL on 5A homologous to the durum wheat grain Cd accumulation *Cdu1* locus and homologous rice loci *OsHMA3* [69], likely co-located with the *Qgcd.tamu.5A.577* identified in this study, which is physically close to the *Vrn-A1* gene. Besides grain Cd concentration, *Vrn-A1* has been found to be associated with copper tolerance [70]. Wang et al. [10] found wheat grain Cd concentration QTLs on chromosomes 1B, 1D, 4A, 4B, 5A, 6B, and 7B. However, no QTL on 3B was reported before, indicating novel loci associated with Cd concentration identified in this study.

## 5. Conclusions

This study used a RIL population derived from elite lines to detect substantial genetic variations and identified QTLs for wheat end-use quality traits. A total of 209 QTLs in 33 genomic regions were detected for these traits, in which 22 consistent QTLs in nine genomic regions were for kernel quality and dough rheological traits and five were QTLs associated with grain mineral element concentrations. Based on the physical locations, some QTLs were close to the loci that were previously well studied or known genes for function associated with end-use quality. Novel and stable QTLs for dough rheological traits were reported in this study on 1D between 64 to 66 Mbp and on 5B at 424 Mbp. QTLs on these two genomic regions mainly contributed by additive effects for several traits, such as MPI, MPV, MRI, and MRS on 1D and MLI and MPT on 5B. The SNP markers closely linked to QTL can be used in designing KASP markers to accelerate the improvement of wheat end-use quality.

**Supplementary Materials:** The following are available online at https://www.mdpi.com/article/10.3390/agronomy11122519/s1, Figure S1: Boxplot analysis of end-use quality traits. Traits included are: (a) Flour protein content, (b) Hardness index, (c) Kernel diameter, (d) Midline left integral, (e) Midline left slope, (f) Midline left time, (g) Midline left value, (h) Midline left width, (i) Mid-

line mixing stability or tolerance, (j) Midline peak integral, (k) Midline peak time, (l) Midline peak value, m) Midline peak width, (n) Midline right integral, (o) Midline right slope, (p) Midline right time, (q) Midline right value, (r) Midline right width, (s) Midline tail integral, (t) Midline tail slope, (u) Midline tail value, (v) Midline tail width, w) Midline time_X integral, (x) Midline time_X slope, (y) Midline time_X value, z) Midline time_X width, (aa) Single kernel weight, (ab) Water absorption, (ac) As concentration, (ac) As concentration, (ad) Ca concentration, (ae) Cd concentration, (af) Co concentration, (ag) Cu concentration, (ah) Fe concentration, (ai) K concentration, (aj) Li concentration, (ak) Mg concentration, (ai) Mn concentration, (am) Mo concentration, (an) Na concentration, (ao) Ni concentration, (ap) P concentration, (aq) Rb concentration, (ar) S concentration, (as) Sr concentration, (at) Ti concentration, (au) Zn concentration. X-axis is environment: BS14, ET14, ans HY13. Y-axis represents the corresponding trait value under the respective environments. Descriptive statistics are on top of each boxplot. Figure S2: LOD profile and additive effects of end-use quality QTL detected in the environments (a)BS14, (b) ET14, (c) HY13, and grain mineral concentraiton in environment (d) ET14. Traits are Flour protein content, Hardness index, Kernel diameter, Midline left integral, Midline left slope, Midline left time, Midline left value, Midline left width, Midline mixing stability or tolerance, Midline peak integral, Midline peak time, Midline peak value, Midline peak width, Midline right integral, Midline right slope, Midline right time, Midline right value, Midline right width, Midline tail integral, Midline tail slope, Midline tail value, Midline tail width, Midline time_X integral, Midline time_X slope, Midline time_X value, Midline time_X width, Single kernel weight, Water absorption, As concentration, As concentration, Ca concentration, Cd concentration, Co concentration, Cu concentration, Fe concentration, K concentration, Li concentration, Mg concentration, Mn concentration, Mo concentration, Na concentration, Ni concentration, P concentration, Rb concentration, S concentration, Sr concentration, Ti concentration, Zn concentration. The top panel of each figure shows LOD profile with chromosomal position along the x-axis (cM of 21 chromosomes) and LOD score on the y-axis. The bottom panel of each figure show an additive effect profile with chromosomal position along the x-axis (cM of 21 chromosomes) and additive effect values on the y-axis (values of each trait have different units). Positive additive effects means that the favorable alleles increasing traits were from CO960293-2 while negative values indicate that the favorable alleles increasing the traits were from TAM 111. Figure S3: LOD profile of additive, additive-by-environment detected in the multi-environment QTL analysis for (a) Flour protein content, (b) Hardness index, (c) Kernel diameter, (d) Midline left integral, (e) Midline left slope, (f) Midline left time, (g) Midline left value, (h) Midline left width, (i) Midline mixing stability or tolerance, (j) Midline peak integral, (k) Midline peak time, (l) Midline peak value, (m) Midline peak width, (n) Midline right integral, (o) Midline right slope, (p) Midline right time, (q) Midline right value, (r) Midline right width, (s) Midline tail integral, (t) Midline tail slope, (u) Midline tail value, (v) Midline tail width, (w) Midline time_X integral, (x) Midline time_X slope, (y) Midline time_X value, (z) Midline time_X width, (aa) Single kernel weight, (ab) Water absorption. LOD profile with chromosomal position is shown on the x-axis (cM of 25 LGs) and LOD score on the y-axis. Positive additive effects mean that the favorable alleles increasing traits were from CO960293-2 while negative values indicate that the favorable alleles increasing the traits were from TAM 111. Figure S4: Epistatic interaction for LOD > 5 between QTL for flour protein content, midline left integral, midline left slope, midline left time, midline left value, midline left width, midline mixing stability or tolerance, midline peak integral, midline peak time, midline peak value, midline peak width, midline right integral, midline right slope, midline right time, midline right value, midline right width, midline tail integral, midline tail slope, midline tail value, midline tail width, midline time_X integral, midline time_X value, midline time_X width, single kernel weight, water absorption. The numbers on the rings represent the peak cM position of the on chromosomes and numbers on each line show the total LOD score of that epistasis effects. The detailed information were listed in Table S7. Table S1. Analysis of variance, heritability and mean performance. Table S2. Correlation matrix for kernel characteristics and rheological properties for data averaged across environments. Table S3. Correlation matrix for kernel characteristics and rheological properties for environment BS14. Table S4. Correlation matrix for kernel characteristics, dough rheological properties, and grain mineral concentrations for environment ET14. Table S5. Correlation matrix for kernel characteristics and rheological properties for environment HY13. Table S6. QTL for end-use quality traits detected once from individual and across environment analyses. Table S7. Epistasis and epistasis by environment interactional effects of end-use quality traits.

**Author Contributions:** Conceptualization, S.L. and J.C.R.; validation, S.Y. and S.O.A.; formal analysis, S.Y.; methodology, S.L., A.M.H.I., J.C.R., Q.X. and M.J.G.; investigation, S.Y. and M.J.G.; resources, J.C.R. and Q.X.; data acquisition and interpretation, S.O.A., J.M.A., A.M.H.I., J.C.R., Q.X., M.J.G., G.Z., J.A.B., K.E.J. and S.L.; data curation, S.Y.; writing—original draft preparation, S.Y.; writing—review and editing, G.Z., M.J.G., J.M.A. and S.L.; visualization, S.Y.; software, S.Y.; supervision, S.L. All authors have read and agreed to the published version of the manuscript.

**Funding:** The funding for this work was from the Monsanto Beachell-Borlaug International Scholars (MBBIS) Program, Texas Wheat Producers Board, Texas A&M AgriLife Research, USDA-ARS Triticeae Coordinated Agricultural Project (TCAP) 2015-67012-25178, and National Institute of Food and Agriculture, Award Numbers: 2017-67007-25939 and 2019-67013-29172.

**Data Availability Statement:** The data that support the findings of this study are available from the corresponding author, S.L., upon reasonable request.

**Acknowledgments:** We are grateful to Shiaoman Chao and the staff at the USDA-ARS genotyping center, Fargo, ND for their help in the genotyping work. We acknowledge the help of technicians Hangjin Yu, Sharis Vader, Lisa Garza, and Maria Fuentealba and undergraduate students who helped in processing samples and quality analysis.

**Conflicts of Interest:** The authors declare no conflict of interest.

## Abbreviations

| | |
|---|---|
| ANOVA | analysis of variance |
| CTAB | cetyltrimethylammonium bromide |
| FPC | flour protein concentration |
| GCD | grain Cd concentration |
| GCO | grain Co concentration |
| GMO | grain Mo concentration |
| HDI | hardness index |
| HMW-GS | high molecular weight glutenin sub-unit |
| ICIM | inclusive composite interval mapping |
| KASP | Kompetitive Allele Specific PCR |
| KD | kernel diameter |
| MLI | midline left integral |
| MLS | midline left slope |
| MLT | midline left time |
| MLV | midline left value |
| MLW | midline left width |
| MMST | midline mixing stability or tolerance |
| MPI | midline peak integral |
| MPT | midline peak time |
| MPV | midline peak value |
| MPW | midline peak width |
| MRI | midline right integral |
| MRS | midline right slope |
| MRT | midline right time |
| MRV | midline right value |
| MRW | midline right width |
| MTI | midline tail integral |
| MTS | midline tail slope |
| MTV | midline tail value |
| MTW | midline tail width |
| MTX | midline time_X integral |
| MTXS | midline time_X slope |
| MTXV | midline time _X value |
| MTXW | midline time _X width |
| QTL | quantitative trait loci |
| RIL | recombinant inbred line |

SKW      single kernel weight
WAB     water absorption

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
