# Peer review of "Genetic Mapping of Quantitative Trait Loci for End-Use Quality and Grain Minerals in Hard Red Winter Wheat"

_agronomy, doi:10.3390/agronomy11122519_

Round 1

Reviewer 1 Report

Comments to manuscript Agronomy 1464035

Manuscript title:

Genetic Mapping of End-Use Quality Quantitative Trait Loci in 2 Hard Red Winter Wheat

The manuscript presents research on a qtl analysis of quality related traits in red winter wheat. The basic approaches are appropriate, the methodology and results are clearly explained.

I suggest that authors restructure the introduction in order to have a better connection with the context in different paragraphs.

There are few comments/concerns which comes in details below:

Have authors performed any normalization/standardization of data across all environments?

In the methodology, the number of replications for phenotyping and quality evaluation needs to be added. Moreover, the statistical model needs to be described.

Lines 59-64: The good logical connection between these lines and above lines are missing!

Line 153: …where σGR2 is the …

Line 192: I suggest to have a (suppl.) table to summarize the results of heritabilities.

Line 323: the sentence is vague!

Line 395: ... in wheat breeding program.

References need to be checked, e.g. line 584, 602, 660 (bold 2008), 720 (paper title?) and add the doi for all refernces if available.

Tables and Figures:

I suggest to have at least 1 figure in the main body of the paper rather than supplementary file.

Table 1 needs to be accurately adjusted in order to be better understood.

Reviewer 2 Report

Dear Editor

The manuscript titled "Genetic Mapping of End-Use Quality Quantitative Trait Loci in 2 Hard Red Winter Wheat" is good attempt and could be important contribution in order to select appropriate genotype for breeding. However, in my opinion, the work requires major corrections as suggested below.

  • Material and methods

- Please mention the experimental design and number of replication.

- Did you test the segregation rate distortion for the markers.

  • Results:

 In the result section please tells because of segregation distortion how many of markers were filtered out?
Please define the origin of the superior additive allele for each QTL?.

  • Discussion needs thoroughly revised by using good thought process rather than just citing other work.

.

Reviewer 3 Report

Please change the colors and add spaces between chromosomes in Supplementary Figure 4. Currently it is completely not colorblind-friendly and hard to understand.   Lines 193-222 are hard to read and perceive. It could be more easy if you make a correlation matrix plot with color legend and put it in the article.   Is there a possibility that pleiotropic QTLs, especially large ones, like 424 Mb on chromosome 5B, will be actually two or more linked QTL and could be identified due to using of larger amount of SNPs? Is there a correlation between number of SNPs in pleiotropic and not pleiotropic QTLs?
